# Effectiveness of Bubble Continuous Positive Airway Pressure (BCPAP) for Treatment of Children Aged 1–59 Months with Severe Pneumonia and Hypoxemia in Ethiopia: A *Pragmatic Cluster Randomized Controlled Clinical Trial*

**DOI:** 10.3390/jcm11174934

**Published:** 2022-08-23

**Authors:** Meseret Gebre, Kassa Haile, Trevor Duke, Md. Tanveer Faruk, Mehnaz Kamal, Md Farhad Kabir, Md. Fakhar Uddin, Muluye Shimelis, Bethelhem Solomon, Abebe Genetu Bayih, Alemseged Abdissa, Taye Tolera Balcha, Rahel Argaw, Asrat Demtse, Abate Yeshidenber, Abayneh Girma, Bitseat W. Haile, Tahmeed Ahmed, John D. Clemens, Mohammod Jobayer Chisti

**Affiliations:** 1Armauer Hansen Research Institute, Addis Ababa 1005, Ethiopia; 2Centre for International Child Health, Royal Children’s Hospital, The University of Melbourne, Melbourne 3125, Australia; 3International Centre for Diarrhoeal Disease Research, Bangladesh (icddr,b), Dhaka 1212, Bangladesh; 4Department of Pediatrics and Child Health, Black Lion Hospital, Addis Ababa 3614, Ethiopia; 5Department of Pediatrics and Child Health, St. Paul’s Millennium Medical College Hospital, Addis Ababa 1271, Ethiopia; 6Department of Pediatrics and Child Health, Yekatit 12 Teaching Hospital, Addis Ababa 1776, Ethiopia; 7International Vaccine Institute, Seoul 08826, Korea; 8Los Angeles Fielding School of Public Health, University of California, Los Angeles, CA 90095, USA

**Keywords:** bubble continuous positive airway pressure, severe pneumonia, hypoxemia

## Abstract

Despite the beneficial effect of bubble continuous positive airway pressure (BCPAP) oxygen therapy for children with severe pneumonia under the supervision of physicians that has been shown in different studies, effectiveness trials in developing country settings where low-flow oxygen therapy is the standard of care are still needed. Thus, the aim of this study is to assess the effectiveness of bubble CPAP oxygen therapy compared to the WHO standard low-flow oxygen therapy among children hospitalized with severe pneumonia and hypoxemia in Ethiopia. This is a cluster randomized controlled trial where six district hospitals are randomized to BCPAP and six to standard WHO low-flow oxygen therapy. The total sample size is 620 per arm. Currently, recruitment of the patients is still ongoing where the management and follow-up of the enrolled patients are performed by general physicians and nurses under the supervision of pediatricians. The primary outcome is treatment failure and main secondary outcome is death. We anticipate to complete enrollment by September 2022 and data analysis followed by manuscript writing by December 2022. Findings will also be disseminated in December 2022. Our study will provide data on the effectiveness of BCPAP in treating childhood severe pneumonia and hypoxemia in a real-world setting.

## 1. Introduction

Although the burden of deaths in children under five years of age due to pneumonia has declined 56% since 2000, still, pneumonia remains the number one infectious cause of under-fives’ deaths, contributing to an estimated 740,000 annual deaths especially in developing countries including sub-Saharan Africa [1]. More than 80% of pneumonia-related deaths occurred in those aged 1–59 months, with the highest burden in the WHO African region [2]. Hypoxemia is found to be the major risk factor for death in children suffering from pneumonia [2]. Thus, appropriate management of hypoxemia may be helpful for a substantial reduction of mortality in childhood severe pneumonia. Despite the provision of oxygen, antibiotics and supportive care, the case fatality rate for children with severe pneumonia and hypoxemia is still high in health facilities of resource-limited settings [3]. In Ethiopia, fifty-one in every one thousand under-five children die before celebrating their fifth birthday [4] and pneumonia accounts for 18% of these deaths [5]. 

Despite its importance in virtually all types of severe acute illness, hypoxemia is often not well recognized or managed in settings where resources are limited [6]. There are many possible clinical options, depending on the availability of medical equipment, in managing a child with severe pneumonia and hypoxemia [6]. In the developed world, in addition to antibiotics and supportive care, the use of high-flow oxygen therapy, humidified high-flow mixture of air and oxygen via a nasal oxygen cannula and continuous positive airway pressure via mechanical ventilator are possible options for treating children with moderate or severe respiratory distress in intensive care units (ICUs) [7].

A systematic review (SR) that identified ten studies [8,9,10,11,12,13,14,15,16,17] and included children with features of severe pneumonia evaluated the efficacy of bubble continuous positive airway pressure (BCPAP) involving 3164 children and revealed a favorable outcome for the use of BCPAP [18]. However, the SR disclosed that most of the studies were conducted in neonates with respiratory distress and there was lack of evidence on the use of BCPAP in childhood severe pneumonia and hypoxemia beyond the neonatal age [18]. It is relevant in this regard that a randomized trial conducted in Bangladesh showed that children who received BCPAP for the treatment of severe pneumonia and hypoxemia had markedly improved outcomes compared with standard low-flow oxygen therapy [19]. Moreover, following the Bangladeshi trial, BCPAP has been used routinely to support nearly 1000, children with severe pneumonia and hypoxemia at the icddr,b hospital in Bangladesh, with beneficial outcomes similar in magnitude to those observed in the trial [20]. Another prospective study, conducted in the Pediatric Emergency Unit of a teaching and referral hospital in North India enrolled 330 children and revealed that nasal BCPAP was safe and effective in children with hypoxemic clinical pneumonia [21]. A sub-analysis from another trial conducted in Ghana found that the use of bubble nasal CPAP was safe and associated with a significant reduction in mortality in infants having undifferentiated respiratory distress [22]. The children in all of these trials were managed and followed up with the direct supervision of physicians.

Even though the beneficial effect of BCPAP for severe pneumonia in children with the supervision of physicians has been shown in different studies, effectiveness trials in developing country settings where low-flow oxygen therapy is the standard of care are still needed to provide evidence for policy on subsequent scale up of this technology in such settings. Therefore, we aimed to conduct a pragmatic cluster randomized effectiveness trial in treating severe pneumonia and hypoxemia among 1–59-month-old children in Ethiopian district hospitals where direct supervision of physicians was available. Before the initiation of the cluster randomized trial, we conducted feasibility and acceptability studies in two tertiary hospitals and two district hospitals to better understand the potential operational challenges for the introduction of bubble CPAP in Ethiopian district hospitals. 

## 2. Materials and Methods

### 2.1. Ethics and Research Approval

This study was approved by ALERT/AHRI Ethics Review Committee (Protocol number: PO/32/17), National Research Ethics Review Committee of Ethiopia and Ethiopian Food and Drug Administration. The trial was registered at ClinicalTrials.gov (ClinicalTrials.gov Identifier: NCT03870243) on 12 March 2019, last update posted 29 October 2021. 

During the initial phase of this study in Ethiopia, another trial conducted in Malawi found that machine-driven bubble CPAP was associated with an increased rate of mortality among high-risk children with severe pneumonia compared with those who received standard-flow oxygen therapy [23]. It is important to note that the children in this study were not monitored or supervised by any physician/pediatrician. Thus, the regulatory body in Ethiopia advised the investigators to conduct this RCT only in those sites where the study patients would be routinely monitored both by physicians and nurses with the supervision of a pediatrician. Consequently, we selected the study hospitals where there is a dedicated room/corner for the study patients in the pediatric ward, and the children are under routine monitoring both by the nurses and physicians supervised by a pediatrician. 

To monitor the overall activity of this trial, a data safety and monitoring board (DSMB) was established, comprising of experts in pediatric emergency care including BCPAP, ethics committee members and a statistician. The results of this study will be disseminated by publication in an international peer-reviewed scientific journal and presented to health policy makers. The study protocol follows the SPIRIT (Standard Protocol Items: Recommendation for Intervention Trials) guidance for protocol reporting (see Table 1 and Appendix A).

The guardian or parent of the study participants must personally sign and date the approved version of the informed consent form before any trial-specific procedures are performed (see Appendix A). Written versions of the participant information and informed consent are presented by study nurse or physician to the participants’ parent or guardian, detailing the exact nature of the trial, what it involves for the participant, known potential side effects, and any risks involved in taking part. It is clearly stated that the participant’s parent or legal guardian is free to withdraw from the trial at any time for any reason without prejudice to future care, without affecting their legal rights, and with no obligation to give the reason for withdrawal.

### 2.2. Study Design

Our study is a pragmatic cluster randomized controlled trial. We randomly assigned 6 district hospitals to BCPAP arm and 6 district hospitals to standard WHO low-flow oxygen therapy arm. An independent researcher matched the clusters using stratified randomization in terms of available facilities such as oxygen supply, availability of ICU, and mechanical ventilation and performed the randomization.

Before initiation of the RCT, a feasibility and acceptability study was conducted as a pilot, initially in 2 tertiary then in 2 district hospitals where we evaluated treatment practices, facilities, and operational challenges for the introduction, clinical use, and maintenance of bubble CPAP. The study in district hospitals was conducted to assess the applicability of bubble CPAP in less intensive facilities in the context of district level hospitals (where there is less manpower and equipment). Children with severe pneumonia and hypoxemia diagnosed by hospital physicians were enrolled consecutively and put on bubble CPAP. During enrolment of the eligible children, we followed the WHO recommended “Management Algorithm of Children 1–59 months with Acute Respiratory Infections”. In addition, we also performed a qualitative study which included key informant interviews, in-depth interviews and focus group discussions with nurses, physicians and parents/guardians of children who received BCPAP and observation of children while on BCPAP. The protocol was revised by addressing the observed operational challenges before the initiation of the RCT.

### 2.3. Study Setting

The cluster RCT is being performed in 12 district hospitals. All district hospitals in Ethiopia have general practitioners but not all have the availability of pediatricians. The characteristics of the hospitals included in our study is similar in terms of most of the facilities (Appendix A). The current practice of oxygen delivery methods in all hospitals is via low-flow (LF) oxygen therapy and remote use of locally constructed bubble CPAP for selected patients in a few hospitals based on clinicians’ decisions, especially those who fail with LF. Children with severe pneumonia are also placed on IV antibiotics and given other supportive care including maintenance fluids. Children are placed on mechanical ventilators if they have any specific indication according to a standard guideline followed in study hospitals.

### 2.4. Study Participants

Children aged 1 to 59 months are eligible for study enrolment upon meeting the clinical criteria for severe pneumonia, as defined by the WHO classification updated in 2013 [24]. Those children who fulfil WHO criteria for severe pneumonia are screened by the study staff for eligibility. All screened patients are allotted a unique identification number and assessed for eligibility. Eligibility criteria included age between 1 month and 59 months and fulfillment of WHO clinical criteria for severe pneumonia (these include a child with cough or difficulty in breathing and any of the danger signs which are: hypoxemia (oxygen saturation < 90% in room air) or central cyanosis, grunting respiration, or inability to breastfeed or drink, lethargy or reduced level of consciousness, or convulsions). All children had to have hypoxemia and written informed consent from the parent/guardian to be included in the study. Children are excluded if they have any one of the following: known congenital heart disease, asthma, or upper-airway obstruction, tracheostomy, pneumothorax or requirement for mechanical ventilation for any specific reason as decided by the clinician.

### 2.5. Treatment Procedures

Hospital standard-of-care management is initiated as soon as a patient is admitted. However, the study-specific intervention starts following ascertainment of eligibility, acquisition of informed consent and enrolment. The flowchart for study procedures is given in Figure 1. In addition to the prevailing standard of care in the hospitals, we used a color-coded pediatric monitoring and response chart that helps to identify deteriorating children to prompt appropriate actions. 

After ensuring that inclusion and exclusion criteria are met, the parent/guardian is provided with the consent form for their reading. If the parent/guardian is unable to read, the physician/nurse reads the consent form with the presence of a witness and requests written informed consent, and the child is enrolled after obtaining consent. According to the cluster allocation, the study children were put in the BCPAP or WHO LF oxygen therapy. In the BCPAP clusters, we use BCPAP that is locally constructed in Bangladesh and constitutes a BCPAP bottle (filled with water during its use for the patient), tubing from the oxygen source up to the water filled plastic bottle, an oxygen source with flow rate of 5–10 L/min, and a nasal prong for nasal interface to be attached to the child (Appendix A). We start BCPAP at 5 cm depth and 5 L/min oxygen. However, if the desired SpO_2_ is not achieved, we titrate both the depth of BCPAP and oxygen flow every five minutes (1 cm and 1 L/min each time up to a maximum of 10 cm and 10 L/min oxygen flow) with a target SpO_2_ of ≥ 90%. In the LF oxygen arm, flow is started at 1 L/min and increased every 5 min up to a maximum of 2 L/min for children 1 month to 2 years and 4 L/min for above 2 years of age. Follow-up is routinely conducted at first hour and fourth hour of intervention, every 4 h in the first 24 h, then every 8 h till final outcome. Additionally, if there are features of treatment failure at any time point of intervention, further follow up is performed. Additionally, we will provide a colored pediatric pneumonia monitoring and response chart in all study hospitals where the study personnel will record all vital signs, SaO_2_, capillary refilling time, AVPU, pain score and blood glucose. They will use this chart to record information in addition to the study CRF.

### 2.6. Data Collection

A structured case report form is used to collect data which include socio-demographic information (age, sex, monthly income, maternal education, family size) and clinical data (immunization status, breast feeding in the first 6 months, duration of cough, fever, respiratory difficulty and inability to feed, convulsion, history of pneumonia in the last one year, and presence of other additional symptoms). Data collection on detailed clinical examination includes vital signs, anthropometric measurements, mentation, hydration status, respiratory effort, baseline oxygen saturation, chest and cardiac findings, as well as other abnormal physical findings.

In addition, data are collected regarding time from patient arrival to evaluation by physician, diagnoses other than pneumonia, antibiotics prescribed, timing of the first dose of antibiotic, history of prior antibiotic treatment, and duration of oxygen therapy prior to enrollment. After enrollment patients’ clinical data is recorded 1 h after study intervention then every 4 h in the first 24 h followed by every 8 h until the final outcome, which includes outcome after withdrawal of patient from the study or even after LAMA (left against medical advice).

### 2.7. Reporting of Adverse Events

All serious adverse events after initiation of the study are recorded and reported to all IRBs, the Ethiopian Food and Drug Administration, and the study DSMB within 24 h of the event. The serious adverse event report includes a structured template of the patient’s detailed medical history, the study intervention, the sequence of events, and management attempts until the outcome of the patient. For patients who are referred, the outcome is collected from the referral hospital physicians over the phone. 

### 2.8. Outcome Measures

#### 2.8.1. Primary Outcome

The primary outcome of the trial is a composite treatment failure variable, defined as any of the following criteria being met:

A.Presence of severe hypoxemia (SpO_2_ < 85%) at any time after at least one hour of intervention plus signs of respiratory distress such as fast breathing, nasal flaring, chest indrawing, head nodding, cyanosis, grunting respiration or apnea while the child is receiving BCPAP/LF OR,B.Development of an indication for mechanical ventilation when the child is receiving BCPAP/LF OR,C.Death during hospitalization or within 72 h of LAMA OR,D.LAMA while still on study intervention (BCPAP or LF oxygen therapy)

All LAMA cases will be contacted by site study staff and reconfirmed by PI or a PI’s delegate via phone or home visit to identify the final outcome of each child.

#### 2.8.2. Secondary Outcomes

Death during hospitalizationDeath during hospitalization plus within 72 h of LAMAPneumothorax Aspiration pneumoniaLength of hospital stayIncidence of nasal trauma, gastric distention, shock, or air leaksDuration of BCPAPTime from decision to start BCPAP to time of start of BCPAP

### 2.9. Sample Size

We calculated our recruitment goal which is a total of 1240 children (620 children each for LF oxygen and bubble CPAP therapy) for a power of 80% and effect size of 50% reduction in the risk of treatment failure among children enrolled in the BCPAP group compared to the LF group (anticipating a relative reduction of treatment failure from 10% to 5% as from chart analysis we found 10% treatment failure among low-flow facilities in 2019) for 12 clusters (103–104 patients in each cluster), adjusted for intra-cluster correlation coefficient (ICC) at a level of 0.003 required for cluster randomization. For the calculation of sample size, we used a formula where, DF, Design effect = 1 + δ(m − 1) (here DF = 1.52),

n_1_ = required minimum sample size per arm, δ = the inter cluster correlation, m = the number of patients in each cluster (103.3 in this study).

P_1_ = the percentage of treatment failure of children with pneumonia receiving standard treatment (low flow) = 10% (we have observed an anecdotal 10% treatment failure from available retrospective data in 12 district hospitals). 

P_2_ = the expected percentage of treatment failure of children with pneumonia who will receive intervention (bubble CPAP) = 5% (we have observed 5% treatment failure in two district hospitals during the feasibility and acceptability stage of our study which correspond 50% effect size), Z_α_ = the z-score value (here, Z_α_ = 1.96 at 5% level of significance), Z_β_ = the z-score value (here, Z_β_ = 0.84 at 80% power).

We are intending to enroll same number of children (103–104) having our eligibility criteria from each hospital (cluster) irrespective of receiving intervention (bubble CPAP) or standard treatment (low flow).

### 2.10. Data Analysis

For all enrolled children, data will be analyzed using STATA-15. In simple analyses, intergroup comparisons of categorical variables will be statistically assessed with the χ^2^ test, or Fisher’s exact test when assumptions for the χ^2^ test are not fulfilled. Comparisons of continuous variables will be analyzed using the independent sample t-test, or the Mann–Whitney test when parametric assumptions are not met. Primary and secondary outcomes will be compared by calculating relative risks (RRs) and their 95% confidence intervals using bi-variate analysis. Log-linear binomial regression will be applied to assess relative risks for the association between treatment group and primary and secondary outcomes after adjusting for potential confounders, if any. Variables other than primary and secondary outcomes, significantly associated with treatment groups, will be considered as the confounders. We will also adjust the potential design effect of the clusters during the regression analysis.

Before the enrolment of the patients in the study, with the permission of the DSMB we conducted a run-in period (to make the study staff familiar with the study procedure) where we have enrolled 2 participants in each hospital and none of them will be included in the analysis. 

### 2.11. Data Quality Assurance

#### 2.11.1. Clinical Supervision

For each patient, the study site supervisors follow the enrollment process and make sure that appropriate clinical follow-up is conducted for the study participants and data collection tools are filled appropriately.

#### 2.11.2. Data Management and Monitoring

An unannounced supervision visit is made to each study site every one to two months by the principal investigator and the co-investigators to assess compliance with study protocol procedures. There is also a weekly data quality check by the investigator team to resolve any issues immediately. In addition, a clinical internal monitor visits the study sites every two months to assure data completeness and quality, which is followed by feedback to the PI and investigators and also a follow-up to ensure issues are resolved. The DSMB also sits at least once after every 6 months and at times depending on the need to know the study progress and data procedure.

## 3. Study Progress

Enrollment into the trial started on 5 April 2021 and is ongoing. We anticipate to complete enrolment by September 2022 and data cleaning, analysis, report writing, and dissemination to be underway by December 2022. Enrollment into the feasibility and acceptability study was started on September 2019 and was completed in July 2020.

## 4. Discussion

The purpose of this clinical trial is to determine whether BCPAP is able to reduce the risk of treatment failure in children with severe pneumonia and hypoxemia compared with WHO standard low-flow oxygen therapy. Despite the provision of WHO recommended supportive care, antibiotics and low-flow oxygen therapy, there is still excess mortality in children with severe pneumonia and hypoxemia in developing countries.

The provision of mechanical ventilation for hypoxemic children in developing countries through endotracheal intubation is not available, feasible, or affordable, except in certain referral hospitals [14]. On the other hand, in Western countries an increasingly used method of respiratory support beyond standard flow oxygen therapy uses a humidified high-flow mixture of air and oxygen via a nasal oxygen cannula (HFNC) for the treatment of bronchiolitis in order to reduce the need for mechanical ventilation [9]. However, a meta-analysis revealed that BCPAP oxygen therapy was associated with a lower risk of treatment failure than HFNC in infants aged 1–6 months with ALRI, moderate-to-severe respiratory distress, and severe hypoxemia. No differences were found in intubation and mortality between HFNC and standard oxygen therapy or BCPAP [25]. Moreover, a recent finding from a different single-center study showed that oxygen therapy delivered by ultra-low-cost bubble CPAP is a possible alternative with beneficial results [21].

In countries like Ethiopia, where low-flow oxygen therapy is the only respiratory support available for most of the children with severe pneumonia and hypoxemia, having simple, inexpensive, and effective methods of providing additional respiratory support could substantially reduce treatment failure and deaths from pneumonia. Importantly, we conducted a study of the feasibility and acceptability of bubble CPAP in Ethiopian hospitals, and this helped us to design our cluster randomized trial. These findings should also help in understanding the benefit of BCPAP oxygen therapy in Ethiopian settings where the availability of ventilators and other respiratory support devices and high caliber professionals are scarce despite providing care for quite a number of children with severe pneumonia and hypoxemia. 

The use of traditional RCTs may produce findings that may not generalize the setting(s) for the trials. The RCT of BCPAP in Bangladesh demonstrates a significant reduction in mortality and treatment failure among Bangladeshi children having severe pneumonia and hypoxemia, but was a single-center trial in an urban hospital, and the trial was conducted with intensive monitoring and additional study personnel [19]. Policy makers may question the applicability of these results to many real-life clinical settings, especially settings where manpower and monitoring are less. Although a recent trial of CPAP in a larger sample in Ghana supported the Bangladesh study [22], a trial conducted in Malawi found that machine-driven bubble CPAP was associated with an increased rate of mortality among high-risk children with severe pneumonia compared with those who received standard-flow oxygen therapy [23]. There were, however, differences in the patient populations, bubble CPAP devices, treatment plans, and monitoring of the patients among these three studies. The Malawian study included a significant number of children with clinical features that were excluded in the Bangladesh and Ghana studies. Moreover, the treatment given to the Malawian study children was not monitored or supervised by a physician/pediatrician. This feature prompted the recommendation by the Ethiopian Food and Drug Authority (EFDA) in Ethiopia that all of the study hospitals selected for our study not only had physicians administering care, but also supervision by a pediatrician, and that all study subjects be managed in a dedicated room/corner under direct vigilance of the nurses in the pediatric ward. If the results of this effectiveness trial on the clinical use of locally made low-cost BCPAP is found to be beneficial, it could be implemented in other developing country settings where the supervision and routine monitoring of the patients by physicians headed by a pediatrician are available. However, the supervision of pediatricians in most district hospitals in other developing country settings might not be available. Therefore, more evidence may still be required in the future on implementation of BCPAP in frontline hospitals that lack pediatricians in Ethiopia.

## Figures and Tables

**Figure 1 jcm-11-04934-f001:**
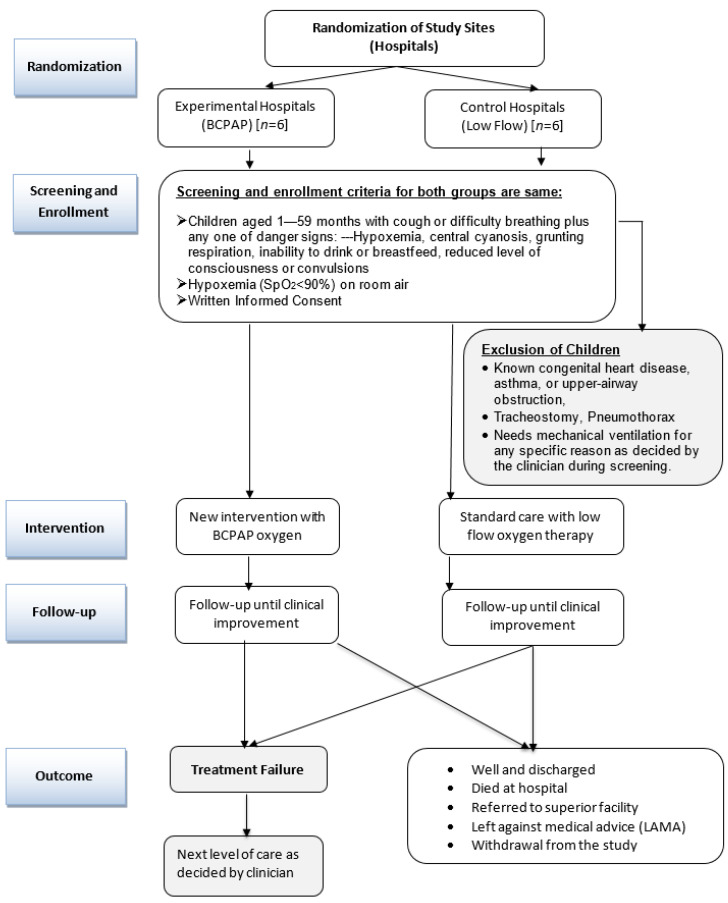
Flowchart for the pragmatic cluster randomized controlled trial.

**Table 1 jcm-11-04934-t001:** Standard Protocol Items: Recommendation for Interventional Trials (SPIRIT) figure showing the schedule of enrollment, interventions, and assessments.

	STUDY PERIOD					
	Screening	Enrolment	Intervention	Follow Up		
TIMEPOINT	Day 0	Day 0	Day 0	Day 0 to Final Outcome in Hospital		
**ENROLMENT:**						
Eligibility screen	X					
Informed consent	X					
Allocation		X				
**INTERVENTIONS:**						
BCPAP			X	X	X	X
Low flow			X	X	X	X
**ASSESSMENTS:**						
Demographics		X				
Clinical variables		X		X	X	X
**PRIMARY OUTCOME**						
Treatment failure				X	X	X
**SECONDARY OUTCOMES**						
Death				X	X	X
Pneumothorax				X	X	X
Aspiration Pneumonia				X	X	X

‘X’ denotes for the schedule of enrollment, interventions, and assessments.

## Data Availability

Not applicable.

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
