# Peer review of "Effectiveness of Bubble Continuous Positive Airway Pressure (BCPAP) for Treatment of Children Aged 1–59 Months with Severe Pneumonia and Hypoxemia in Ethiopia: A Pragmatic Cluster Randomized Controlled Clinical Trial"

_jcm, 2022, doi:10.3390/jcm11174934_

Round 1
Reviewer 1 Report
This is a very important study that will contribute towards the evidence we need to ensure optimal oxygen therapy for children with pneumonia in real world clinical settings.
It would be good for the authors to describe the study setting in more details. They indicate that the studies are done at district hospitals, but the definition of district hospital may vary from country to country in terms of scope of services, equipment, and human resource. The readers may already want to find out how the study setting compares to their own settings.
The authors indicate that this is a real-world setting study, but this may not be true especially low-income settings. For example, children with severe pneumonia are not always seen by a physician/doctor who is also supervised by a Paediatrician. The intensity of the monitoring in this study is also not possible even in some tertiary care facilities. I would argue that the study setting and the procedures in the study are near real-world.
Is the standard of routine care the same across all sites? The authors mention matching the sites, but the details are not provided but this would be helpful in understanding the baseline characteristics of the different sites
Author Response
Thank you for your valuable suggestions. We tried our level best to address all the issues raised by you

Reviewer 2 Report
The investigators in this protocol of a pragmatic cluster randomized controlled trial study the effectiveness of bubble CPAP in children (1-59 months) with severe pneumonia and hypoxemia in Ethiopia.
There have been few single center RCTs published around this subject which showed that bCPAP with physician oversight improves outcomes compared to low flow oxygen. However, a trial in Malawi without adequate monitoring showed the opposite results. So, it is imperative to study the barriers and facilitators of bCPAP as well as identify patients who would best benefit from this intervention. Investigators in this trial aim to take this further and intend to do RCT in real world (multicenter center study, with better monitoring and physicians’ oversight) in Ethiopia. After doing a feasibility and acceptability pilot study (which involved qualitative components; in depth interviews and FGD), they will implement bCPAP in 12 district hospitals which will have a pediatrician’s oversight. WHO criteria for diagnosis of severe pneumonia will be used to enroll children, excluding those with CHD, Asthma, upper airway obstruction, those with tracheostomy, pneumothorax, and those requiring invasive mechanical ventilation at arrival. Primary outcome is treatment failure ( a composite variable) defined as 1) persistence of hypoxemia (SpO2<85%) or signs of respiratory distress at one hour after the start of bCPAP OR 2)development of an indication of Mechanical ventilation while on bCPAP, OR 3) Death during hospitalization or within 72 hours of LAMA OR 4) LAMA while still on study intervention. They intend to recruit 1240 children (620 in each arm).
The overall study questions, study design, conduct of trial, ethics and analysis plan are well presented and clear. However, addressing following points will make the protocol more useful and valuable.
1. Since the feasibility and acceptability study has been done before the trial was design, do investigators intend to publish results? This would answer some key questions around implementation of bCPAP in low resource setting. Specially mentioning operational challenges encountered and how were they handled? This will have implications for its implementation at other sites too?
2. How undiagnosed CHD or other chronic lung conditions will be differentiated from pneumonia? Children coming for the first time might fulfill the WHO criteria of severe pneumonia without having pneumonia, what procedure will be followed to include /exclude them?
3. It would be good to list a basic detail of the study hospitals; beds, oxygen sources, government or private sectors, other resources, availability of pulse oximeters, patients: nurse ratio, patient: physician ratio, usual pneumonia care in these hospitals (cxr, antibiotics etc), availability of mechanical ventilator, triage process.
4. I would suggest adding more details around monitoring. The authors state that data will be collected after 1 hour of intervention, but monitoring will be done every 5 minutes in case improvement is not noted. Will this data be documented? The case report form doesn’t show this. Similarly in previous studies many patients had septic shock and heart failure, which makes monitoring more important, specially monitoring of intake and output, vitals signs and perfusion. Do investigators think this aspect is of critical importance and can be looked at in more details.
5. Since bCPAP can be a time sensitive intervention, do investigators intend to monitor time from decision to start bCPAP to time of start of bCPAP (delay may be happen because of limited staffing, delayed consent, etc)? This will bring important data to understand and make changes for its wider implementation.
6. How the seal of bCPAP be ensured and how frequently circuit will be assessed? Will water bottle be changed during the entire course of treatment and if yes how frequently? Do authors anticipate any risk of nosocomial infection through this circuit?
Author Response
Thank you for your valuable suggestions. We tried our best to address all the issues raised by you.

Reviewer 3 Report
This protocol focused on the BCPAP and standard low flow oxygen therapy, aimed at versus the effectiveness between two therapies in developing countries. This protocol’s expected study design reasonable and study sample were large, still existing some small flaws needs to be corrected listing as follows.
1. The first phrase of introduction part was too long to illustrate pneumonia and hypoxemia, authors are advised to write concisely.
2. Figures in the protocol were vague and influence reading, the author should change to higher resolution pictures.
3. 2.8 outcome measures need adjust the format of A, B, C, D. Four spaces look abrupt.
4. Did the ethics section have an ethics number?
Author Response

(The authors gave the same response as above.)
